# Emotion Recognition: Photoplethysmography and Electrocardiography in Comparison

**DOI:** 10.3390/bios12100811

**Published:** 2022-09-30

**Authors:** Sergio Rinella, Simona Massimino, Piero Giorgio Fallica, Alberto Giacobbe, Nicola Donato, Marinella Coco, Giovanni Neri, Rosalba Parenti, Vincenzo Perciavalle, Sabrina Conoci

**Affiliations:** 1Department of Educational Sciences, University of Catania, via Biblioteca 4, 95124 Catania, Italy; 2Department of Biomedical and Biotechnological Sciences, Section of Physiology, University of Catania, via S. Sofia 89, 95125 Catania, Italy; 3INSTM (National Interuniversity Consortium of Science and Technology of Materials), via G. Giusti 9, 50121 Firenze, Italy; 4Department of Engineering, University of Messina, Contrada Di Dio, 98158 Messina, Italy; 5Department of Sciences of Life, Kore University of Enna, Cittadella Universitaria, 94100 Enna, Italy; 6Department of Chemical, Biological, Pharmaceutical and Environmental Science, University of Messina, Viale F. Stagno d’Alcontres 31, Vill. S. Agata, 98166 Messina, Italy; 7LAB Sense Beyond Nano—URT Department of Sciences Physics and Technologies of Matter (DSFTM) CNR, Viale F. Stagno d’Alcontres 31, Vill. S. Agata, 98166 Messina, Italy; 8Department of Chemistry ‘‘Giacomo Ciamician’’, University of Bologna, Via Selmi 2, 40126 Bologna, Italy; 9Istituto per la Microelettronica e Microsistemi, Consiglio Nazionale delle Ricerche (CNR-IMM), Strada VIII n. 5, 95121 Catania, Italy

**Keywords:** bio-signal processing, photoplethysmography, unobtrusive sensing

## Abstract

Automatically recognizing negative emotions, such as anger or stress, and also positive ones, such as euphoria, can contribute to improving well-being. In real-life, emotion recognition is a difficult task since many of the technologies used for this purpose in both laboratory and clinic environments, such as electroencephalography (EEG) and electrocardiography (ECG), cannot realistically be used. Photoplethysmography (PPG) is a non-invasive technology that can be easily integrated into wearable sensors. This paper focuses on the comparison between PPG and ECG concerning their efficacy in detecting the psychophysical and affective states of the subjects. It has been confirmed that the levels of accuracy in the recognition of affective variables obtained by PPG technology are comparable to those achievable with the more traditional ECG technology. Moreover, the affective psychological condition of the participants (anxiety and mood levels) may influence the psychophysiological responses recorded during the experimental tests.

## 1. Introduction

The steady development of accurate emotion recognition techniques allows for their application in many fields including marketing, robotics, psychiatry, entertainment, and game industries [1,2,3,4,5]. Some of the technologies used to detect emotion can also be used as biofeedback techniques to counter or mitigate stress [6].

The mental state of a person can be investigated with sensors measuring physiological parameters coming from different parts of the body. Firstly, there are methods based on the analysis of signals collected directly from the Central Nervous System by both electrical and optical transduction. An extensive review of emotion recognition research based on Electroencephalography (EEG) and functional Near Infrared Spectroscopy (fNIRS) is reported in the literature [7,8,9]. Sensors capable of extracting information from facial expression, tone of voice, and posture are the most obvious and natural choice at first sight; however, they have an important drawback: these modes of expression are largely under the voluntary control of the individual, therefore they can, to a certain extent, be modified and thus mask the real emotional state.

On the contrary, sensors that obtain information from organs that are under the control of the Autonomous Nervous System (ANS) are more reliable, since any intentionality of reflexes can be excluded. The effects of the emotional states, mediated by ANS on the cardiovascular system, the respiratory system, and electrodermal activity have been studied very extensively: in some recent reviews, hundreds of scientific articles have been mentioned and classified [10,11,12].

Most of the instruments used for the recognition of emotions were originally developed for clinical diagnosis purposes and therefore do not have specifications suitable for field use, in real life and for prolonged acquisition, and are often less versatile outside of a controlled laboratory environment.

A clear aim of emotion recognition systems is to be applicable in everyday life [13]. Hence, a major goal is to use sensor setups that are minimally invasive and provoke only minor limitations to the mobility of the user. Consumer wearables devices already offer activity recognition, and recently the first generation of affect (e.g., stress) recognition systems entered the sector [14]. Consumer wearable devices generally use the following signals (in the following order): (a) cardiac cycle (e.g., ECG or PPG), (b) electrodermal activity, (c) respiration, and (d) skin temperature.

The recognition of emotions in real life imposes several limitations on the range of transduction modalities that can be used in practice. In order not to interfere with daily activities, it is necessary to have non-intrusive sensors, i.e., sensors whose presence is so discreet that their presence can be forgotten by the user. The set of non-invasive physiological sensors is not very large. Among these, PPG sensors are the most explored due to their advantages in miniaturization and noninvasiveness [15,16,17]. If the user is seated at a desk, one can use head movement tracking systems based on visible light cameras or face temperature detection systems based on infrared light cameras [18]. If the user is on the move, they can be used for PPG heart rhythm sensors and little else.

In this paper, we present a study about the detection of some physiological parameters related to heartbeat variability using a miniaturized PPG sensor and their comparison with those normally extrapolated from the ECG. Additionally, we discuss whether with this set of parameters we can obtain a good classification of emotions with levels of accuracy comparable to those obtained with other systems.

### 1.1. Emotion Models

Effectively stimulating a predetermined emotional response in a sample of human volunteers and then recognizing it is a challenge. The first difficulty consists of the fact that it is not easy to uniquely define what an emotion is.

Plutchik [19] proposed a psycho-evolutionary classification approach for general emotional responses. He created an emotion wheel to illustrate different emotions and first proposed his cone-shaped (3D) model, or wheel model (2D), in 1980 to describe how emotions were related. He suggested eight primary bipolar emotions: joy versus sadness; anger versus fear; trust versus disgust; and surprise versus anticipation. His model connects the idea of a circle of emotions to a color wheel. Like colors, primary emotions can be expressed with different intensities and can mix with each other to form different emotions.

These are based on the physiological reaction that any emotion creates in animals, including humans. Emotions without color represent an emotion that is a mixture of the two primary emotions.

In recent years, a new model has emerged, the so-called Russell circumplex model of emotions (Figure 1), which argues that affective states are attributable to two main neurophysiological systems, one that explains the value of emotion (along a continuum of pleasantness-unpleasantness) and another that refers to the corresponding physiological arousal/activation level [20].

According to this theory, each emotion can be explained as the linear combination of the two dimensions, varying in the valence (positive or negative) and intensity of activation. Joy, for example, is conceptualized as an emotional state characterized by positive valence and a moderate level of arousal. The subsequent cognitive attribution, which allows for the integration of the two dimensions, the underlying physiological experience and the decisive stimulation, finally allows for the identification of the emotion of joy.

Emotions, according to this theory, should be the final product of the complex interaction between cognitions, elaborated in the neocortical structures, and neurophysiological modifications, linked to the valence and activation systems, regulated by subcortical structures [21].

### 1.2. Emotion Elicitation Protocol

Emotions are closely linked to events, situations, and people, which have acquired an affective meaning over the course of a person’s life. It is therefore expected that the more naturalistic the circumstances in which emotions are provoked, the more likely it is that those emotions reflect the normal experience of the subjects. Technology allows us today to capture, in real time and in everyday circumstances, the physiological reactions related to emotions thanks to the wide range of wearable sensors available on the market. However, the study of emotions induced by certain types of actions, such as giving a speech or driving a car, is problematic for another reason. Motor activity can produce artifacts in the recording of physiological parameters, and so the task of laboratory studies is to separate the physiology of emotions from the physiology of actions and motion [22,23].

The choice of stimulus type is critical and should be designed according to the purpose of the research. Some characteristics that distinguish emotional stimuli must be carefully considered: their ecological validity, complexity, and intensity. Ecological validity refers to the ability of the stimulus to provoke an emotional reaction like the real emotional experiences of daily life. From this point of view, watching film clips is preferable to other stimuli, as they provide rich contextual information. 

Watching movies can elicit a wide range of emotional intensities: from neutral to very intense, probably not as strong as those provoked by real-life events, but stronger than those aroused by static stimuli, like pictures or sounds, due to their great similarity to real emotional experiences.

### 1.3. Emotion Prediction

The study of the heart rhythm, which can be obtained from the ECG, is a powerful tool for extracting information on cognitive functions and emotional responses [24]. 

A healthy heart can rapidly adjust its rhythm adapting to sudden physical and psychological challenges in an uncertain and changing environment. Its oscillations are complex and constantly changing, with nonlinear behavior. They reflect the regulation of autonomic balance, blood pressure (BP), gas exchange, gut, heart, and vascular tone. Heart Rate Variability (HRV) consists of changes in the time intervals between heartbeats (inter-beat interval, IBI). The acronym, HRV, does not indicate a single index but a vast family of indices, all derived from the IBI series. Time-domain indices quantify the amount of variability in measurements of the IBI series. Frequency-domain indices estimate the distribution of absolute or relative power into four frequency bands. For frequency-domain analysis, traces were interpolated using cubic-spline interpolation, and the power spectra were obtained using fast Fourier transform (FFT). The absolute and relative powers of very low frequency (VLF; < 0.04 Hz), low-frequency (LF; 0.04–0.14 Hz), and high-frequency (HF; 0.15–0.4 Hz) bands were measured.

Non-linear measurements allow us to quantify the unpredictability of a time series. A non-exhaustive list of HRV indices is shown in Table 1.

We underline that the HRV indices in Table 1, even the non-linear ones, are derived from the series of IBI, that is, only from the duration of the beat, neglecting other information obtainable from the ECG signal. Alternative features can be extracted using other ECG signal-based techniques [25].

The values of the IBI intervals and the HRV indices can also be obtained from the PPG traces. PPG is a low-cost optical technique widely used in medical devices for monitoring oxygen saturation. A PPG probe essentially consists of two elements: a light source that emits light in the visible or near-infrared (NIR) spectral range, where hemoglobin is the main light absorber, and a photodetector that collects light transmitted or backscattered by biological tissue. 

PPG is sensitive to the volumetric modulations of peripheral arteries induced by the propagation of the pulse pressure wave from the heart. PPG measurement can be performed in two main modalities: in transmission modality, the source and the detector are placed on two opposite surfaces of the same body district (finger, earlobe), whereas in back-scattering modality, the source and the detector are placed on the same surface (wrist, ankle). 

The sensors that capture the PPG signal and process psychophysiological information can be integrated into portable devices, such as smartphones [26], or wearable devices, such as an Optical Heart Rate Monitor [27]. These devices have been used for a long time for heart rhythm measurement and, more recently, as a convenient ECG surrogate for non-clinical HRV analysis. They take a continuous measurement of the PPG waveform and are typically based on LEDs that emit green or yellow light. Since green light has a much higher absorption coefficient than IR light, it is absorbed in the most superficial layers of the skin, so OHRMs are sensitive to blood circulation in the capillaries.

Very often, the two techniques (PPG and ECG) are considered and used as two interchangeable means to measure Heart Rate (HR) and HRV. However, on closer inspection, there are several reasons why the variability of the beat recorded with the PPG, which from now on we will call Pulse Rate Variability (PRV), is something substantially different from HRV [28]. In fact, if the average duration of the beats, obtained with the two methods, is necessarily the same, the duration of the individual beats can be quite different. In Figure 2, we show a typical scatter plot of IBI values extracted from an ECG and PPG synchronous acquisition. The plot is derived from one of our measurements. 

The reason is that electrical impulses from the heart are translated into optical signals through a long series of successive physiological processes [29]:The electrical trigger given by the sinoatrial node and then by the atrioventricular node causes the myocardium to contract;the contraction of the ventricle causes an increase in blood pressure in the left ventricle (the pre-ejection period);when the pressure in the ventricle exceeds the pressure in the aorta, the aortic valve opens, so a blood bolus is pushed into the artery;the aorta dilates to accommodate the blood bolus in proportion to its elasticity (which depends on many factors, first on age); the pressure pulse generates a wave (ABP, arterial blood pressure waveform) that propagates along the walls of the arteries at a speed that depends on the diameter of the arteries, the thickness of the walls, their elasticity, and the viscosity of the blood;the wave pulse reaches the body site where we placed the PPG probe and manifests itself as a rhythmic variation in the diameter of the arteries, which is very small in percentage;the change in the diameter of the arteries results in a change in the volume of blood; the variation of the blood volume results in a variation of the transmitted/back diffused light which is then collected by the photodetector.

The relationships between the variables involved (blood pressure and volume, wave propagation times) are complex and depend on several factors, including external ones. As an example, to adequately describe the relationship between the pressure and volume of the arteries, we must also consider the external pressure. The curve that expresses the relationship between pressure and volume has non-linear portions for extreme values (high blood pressure or high probe pressure). It is increasingly difficult to further expand or collapse the vessel after it reaches certain limits. Furthermore, the stiffness of the artery (or its compliance) is not a static parameter.

Dynamic compliance means that vessels are stiffer when their pressure changes quickly (e.g., intra-beat) and more compliant when their pressure changes slowly (e.g., inter-beat). Therefore, a PPG curve can appear dampened relative to the arterial blood pressure waveform (ABP) because the higher-frequency waveform components are lacking [30].

These few hints are enough to clarify that when we talk about the duration of the systolic tract and the diastolic tract in the ECG, ABP, and PPG curves, we are talking about correlated things, but not the same thing. Therefore, caution should be exercised in considering the variability associated with PPG, i.e., PRV, as a good surrogate for HRV. Some authors experimentally compared the PRV and HRV variability parameters [31] and found that the values obtained with the first method roughly coincide with those obtained with the second method only under certain physiological conditions and within certain limits. Although HRV is a major source of PRV, the latter is generated and modulated by many other sources and factors. Thus, the PRV could contain extra useful biomedical information. Therefore, it is worth considering it as a new and distinct biomarker [32]. The question then arises whether PRV is useful or not, especially if it is useful in the field of emotion recognition.

## 2. Materials and Methods

For the reasons stated above (paragraph 1.2), we decided to provoke emotions by asking our volunteers to watch music videos. To obtain the greatest effectiveness of the test, we adopted a video selection procedure that took into account the characteristics of our sample of subjects: young adults of medium-high culture. The procedure to build our video clip dataset was inspired by the methodology present in the work of Koelstra et al. in 2011 [33]. The video clips were selected through the “last.fm” site, which provides a taxonomy based on the emotional reactions of the musical pieces, by means of tags associated with the songs by users from all over the world, through adjectives characterizing emotional states (e.g., sad, happy, etc.). We initially selected 10 video clips for each emotional quadrant (high valence/high arousal—high valence/low arousal—low valence/high arousal—low valence/low arousal), for a total of 40 video clips. We then administered the 40 video clips to a sample of 40 subjects. The subjects were asked to give an opinion of the individual emotional stimuli based on the Self-Assessment Manikin (SAM). SAM is a widely used non-verbal assessment technique: each manikin pictorially represents a different level of a certain emotional state. Nine of these manikins were displayed on the screen with numbers printed below [34,35,36]. Five emotional variables were assessed: valence, arousal, dominance, pleasantness, and familiarity (Figure 3).

The valence scale ranges from unhappy or sad to happy or joyful. The arousal scale ranges from calm or bored to stimulated or excited. The dominance scale ranges from submissive (or “out of control”) to dominant (or “in full control”). A fourth scale investigates the participants’ personal satisfaction levels for the video. This last scale should not be confused with the valence scale. This measure investigates the tastes of the participants, not their feelings. These first four variables were evaluated with a scale of levels ranging from 1 to 9. Moreover, the participants were asked to rate their familiarity with the song on a scale of 1 (“Never heard of before the experiment”) to 5 (“I knew the song very well”).

Subsequently, we selected eight definitive video clips, two for each emotional quadrant, based on the average scores obtained through the SAM, placed at the extremes of the score range in relation to the valence and arousal dimensions. The first group of 40 volunteers only participated in the selection of the eight music video clips, which were used in the actual experiment.

### 2.1. Emotion Elicitation Protocol

The experimental sample was composed of 31 healthy adult subjects of both sexes (14 male and 17 female) of an age group between 18–39 years (mean = 27.3; standard deviation SD = 4.4).

Information was collected on the participants’ general condition, health status, and whether they were taking any drugs or substances that could alter physiological activation levels. The inclusion criteria were possession of a driving license, good health, and age. Volunteers signed informed consent and were also informed of their right to privacy, nonrecognition, and anonymity. They could withdraw from the study at any time. The study was approved by the Human Board Review and Ethical Committee Catania 1 (authorization n. 113/2018/PO, University of Catania, Italy). The study was performed in agreement with the ethical standards of the Helsinki Declaration.

The sample of 40 volunteers had no subjects in common with the sample of 31 volunteers who were subjected to all tests, i.e., also to physiological measures; however, the 40 subjects had similar biographic features with the experimental sample, i.e., age, sex, and instruction level.

Each subject was seated comfortably in an armchair in front of a screen, with one bracelet on the wrist, where the PPG detector was applied. ECG activity was measured by means of three electrodes: two placed on the arms, forming the lead I of Einthoven’s triangle; the third, placed on the right ankle, ensured the grounding. The subjects were asked to view some music videos, used as triggers to activate emotional reactions.

The experimental session took place like this: the 31 subjects were provided with a series of accurate instructions and performed a practical test to familiarize themselves with the computerized system. Each trial consists of the following steps:A basic recording of 30 s, where the subjects were asked to fix a pre-selected neutral image.Listening to and viewing the music video.Self-evaluation of feelings experienced while watching the videos through the administration of the SAM.

Finally, the participants were given some psychological tests. Specifically, they were required to fill in two assessment tests of the affective state (mood, anxiety), such as the “Profile of Mood State” (POMS) [37] and the “State-Trait Anxiety Inventory-Y” (part 1 and 2; STAI-Y) [36]. We decided to evaluate the mood and the levels of anxiety of the participants to monitor the initial psychic condition (trait anxiety, mood state), able to influence the psychophysiological responses recorded during the experimental tests and the impact they had on the participants (state anxiety).

The questionnaires are detailed below:

(a) STAI-Y parts 1 and 2: a test for the assessment of anxiety levels, consisting of two parts, which measure state and trait anxiety levels. State anxiety is understood in contingent terms, of self-perception here and now; trait anxiety refers to the levels of anxiety as a personological, a distinctive trait of the subject [38].

(b) POMS: a test that measures the mood of the subject, as it is perceived. It measures six mood factors (Tension, Fatigue, Confusion, Vigour, Depression, and Aggression) and returns a global index of the subject’s mood (Total Mood Disturbance, or TMD), referable to the last week as lived by the subject [35].

### 2.2. Signal Acquisition and Pre-Processing

The signals were collected employing a homemade multi-channel system. The optical probe was composed of a light source consisting of a couple of Light Emitting Diodes (LED) emitting red and infra-red light (wavelength range centered, respectively at 735 and 940 nm) coupled with a very sensitive photodetector: the Silicon PhotoMultiplier (SiPM) [39]. The probe was inserted inside a cuff exerting an under-diastolic pressure of 60 mmHg. The cuff was placed on the left wrist. The sampling frequency was 1 kHz.

The off-line analysis included: (a) filtering with a fourth-order, band-pass Butterworth digital filter (from 0.5 to 10 Hz); (b) identification of the R wave of the ECG with the Pan–Tompkins method; (c) identification of the instant of the start of the PPG beat; (d) calculation of the IBI sequences. IBIs obtained from ECG and PPG are named RR and PP, respectively.

After filtering, we divided the signals into single beats and discarded spurious beats, then we truncated traces from the initial part to obtain a duration always equal to 120 s.

### 2.3. Dataset

We considered 30 features which derive from the analysis of the beat length (see Table 1), whether it is expressed as the RR distance between the R peaks of the ECG trace or expressed as the PP distance.

Several additional features, deriving from PPG pulse shape analysis were also obtained. The meaning of these additional parameters, described by La Yang and colleagues [40], is explained in Figure 4 and Table 2.

The last nine parameters (shape parameters, SP) were calculated for each pulse, or for each couple of consecutive pulses, when applicable. Our hypothesis is that, like RR or PP, also the variability of the shape parameters is linked to the inputs of the Autonomous Nervous System. Following this hypothesis, we calculated, for each SP parameter, its RMSSD index and its mean value. 

Physiological parameters have great intra and inter-subject variability due to age, sex, motor activity, and time of day. While selecting healthy volunteers who fall into the same age group, the subjects may have physiological parameters with very different resting values. Normalization is primarily an attempt to subtract the effect of this variability not attributable to emotions. We adopted Z-score normalization: we normalized the data for each feature (for each subject separately) by subtracting the mean value of the measured parameters and dividing the result by their standard deviation.

The self-assessment of the first four emotional variables was expressed according to nine levels of intensity. For the fifth variable (familiarity), only five levels were used. The number of volunteers is not so high as to allow a classification with so many levels. Thus, we grouped the nine levels of intensity (the five levels in the case of the familiarity variable) into two classes. Levels from 1 to 5 belong to the first class; levels from 6 to 9 belong to the second class (see Table 3).

The total number of feature vectors (observations) was 248 (31 volunteers times eight video clips). Ten observations were discarded, because of the poor quality of the registration.

### 2.4. Classification

We used for the classification machine-learning algorithms implemented in Matlab R2019b. We experimented with a variety of algorithms; the ones which gave us the best performances are:(a)KNN—an algorithm that provides a prediction based on k training instances nearest to the test instance. We selected k = 10 and Euclidean distance.(b)SVM—an algorithm for building a classifier where the classification function is a hyperplane in the feature space.

To validate the models, we employed a LOSO (leave one subject out) method. According to this method, the algorithm is trained on the data of N-1 subjects (where N = 31), and after, it was validated on the excluded subject. The procedure is repeated N times so that each of the N subsets is used exactly once as test data. The metrics used to evaluate the models are: accuracy, precision, recall, and F1 score, defined by Equations (1)–(4).
(1)$$Accuracy=\frac{Correctly\;predicted\;classes}{Total\;number\;of\;observations}$$
(2)$$Precision=\frac{Correctly\;predicted\;class\;j}{Total\;predictions\;of\;class\;j}$$
(3)$$Recall=\frac{Correctly\;predicted\;class\;j}{Total\;observations\;of\;class\;j}$$
(4)$$F1\;score=\frac{2\cdot \left(Precision\cdot Recall\right)}{Precision+Recall}$$

We used the HRV and PRV features as two separate sets. The third set is formed by shape parameters (SP) and the fourth by SP + PRV. Each of the four sets of parameters went through a selection process to find discriminative features. A sequential forward selection method was used. The features were added one at a time, having as a selection criterion the accuracy value. The entire process was repeated for each psychological variable.

## 3. Results

We first analyzed the SAM self-assessment values and the data on mood obtained from the POMS and STAI-Y questionnaires. We wondered if the mood and anxiety levels of the volunteers could affect the values assigned to the five emotional variables. We, therefore, calculated the correlation between the two sets of data. On the one hand, we have: TMD, the six mood factors of POMS (Tension, Fatigue, Confusion, Vigour, Depression, and Aggression), State, and Trait Anxiety, a total of nine variables. On the other hand, we have the judgments expressed with the SAM technique after viewing the eight video clips, relating to the five emotional variables (Valence, Arousal, Dominance, Pleasantness, and Familiarity), a total of 40 variables. We found some statistically significant correlations, shown in Figure 5.

We then analyzed the non-normalized experimental values of the physiological parameters to verify: (a) if our group of volunteers responded to the stimuli in a homogeneous way; (b) if the HRV and PRV parameters are well aligned. 

We, therefore, compared the changes in the parameters from one subject to another with the changes in the parameters due to viewing the music videos.

In Figure 6, we show two extreme cases. The Heart Rate changes very little in a subject because of our stimuli but changes more from one subject to another. The RMSSD shows big spreads in both intra and inter-subjects. 

We should prefer little variations in inter-subject and big variations in intra-subject, but this is not true, at least in our experiment.

### 3.1. Alignment of HRV and PRV Parameters

The alignment of the HRV and PRV parameters was verified with linear regression (Table 4) and with Bland–Altman plots. 

The best correlation occurs in the case of the HR parameter; the worst correlations are for the four parameters: ApEn, VLF, LF, and HF. The plots of the PRV versus the HRV values clearly show that the observations belong to two distinct populations (Figure 7). We can evaluate this effect also by the Bland–Altman plots (Figure 8 and Figure 9).

The first graph shows the heart rhythm measured by the two methods. As expected, the average value is approximately equal (bias approximately zero). The difference of about one beat that is found in some measures can be easily explained bearing in mind that the acquisition duration is the same for ECG and PPG, but the number of discarded beats is greater in the PPG trace. The other time domain and non-linear parameters show very good agreement, with three exceptions (as expected): VLF, LF, and HF.

In each frequency range, there is a significant percentage of observations (about 40%) in which the value of the parameter extracted from ECG is greater than that extracted from PPG. The observations with this anomalous behavior are the same in the three graphs and come from a sub-group of our volunteers (12 subjects). It seems that for some reason, the reactivity of these subjects to emotional stimuli is different (in a certain sense greater) than most of the volunteers. The two subgroups do not have statistically relevant differences in terms of sex and age. In the plots of pLF and pHF, this anomalous group of observations disappears (Figure 9).

The results we have shown indicate that responses to emotional stimuli are qualitatively and quantitatively different between subjects, despite having chosen a group of volunteers homogeneous in age and health conditions. The correlations shown in Figure 5 seem to indicate that the values attributed by the different volunteers to the emotional variables are in some way influenced by the state of anxiety or the mood in the last week. The spread of intra and inter-subject PRV parameters, the Bland–Altman plots, and the regression plots tell us that the values of some physiological parameters belong to two different populations. The minority population refers to 12 subjects.

Then, we wondered if among the different volunteers, in addition to the obvious differences due to physiology, there are also differences due to the psychological state of the moment or period [41]. Therefore, we correlated the 30 HRV and PRV parameters with the TMD and STAI-Y parameters. The statistically significant correlation cases are shown in Figure 10. 

We can conclude that there is a small effect of the subject’s mood on some HRV and PRV parameters. An effect that adds up and interferes with the effect caused by watching music videos. 

We underline the fact that while the correlation of Figure 5 is between two sets of self-assessment scores, the correlation of Figure 10 is between one set of self-assessment values and some physiological features.

Finally, we have investigated if, with the PPG-derived parameters, we can obtain a better prediction of emotive variables. Dividing the observations into the classes they belong to (positive or negative, high or low, etc.), we obtain distributions of HRV and PRV features which, in general, have similar *p*-values (see Table 5).

### 3.2. Classification

Our aim was to use a straightforward algorithm to compare parameters extracted from the ECG and PPG waveforms. Thus, we used two of the most popular algorithms in the field of emotion recognition.

In Table 6 and Table 7, we show the results of the classification obtained with two algorithms (KNN and SVM) and with subject-independent LOSO cross-validation. The F1 score and accuracy in both cases are slightly better using PRV features. The classification with PPG shape parameters (SP) obtains accuracy values like those that use HRV or PRV features. This is the most important result of our study. It can be deduced that the autonomic nervous system, in response to emotional stimuli, modulates in a distinctive way not only the duration of the beat but also the shape of the PPG signal.

In Table 8, we report the features we chose with the SFS (Sequential Forward Selection) procedure. We performed a total of 40 classifications (four sets of features times two algorithms times five emotional variables). In many cases, the SFS procedure has selected similar HRV and PRV parameters for the same emotional variable. 

To better appreciate the performance of the KNN and SVM algorithms, in Figure 11 and Figure 12, we make an equal comparison of the classification results obtained with the two methods. The graphs show the accuracy values and the standard deviation of the accuracy (the same reported in Table 6 and Table 7). In total, we have 40 values: 5 emotional variables times, 4 sets of features (HRV, PRV, SP, PRV + SP) times, and 2 algorithms. The standard deviation of the accuracy was calculated based on the accuracy values of each volunteer, i.e., based on the percentage of exact predictions referred to that volunteer.

The SVM algorithm gives us better results than the KNN algorithm; in most cases, we have higher accuracy values and a smaller standard deviation.

### 3.3. Impact on Classification of the Individual Differences

We made an analysis to understand if individual differences produce a disturbing effect that could worsen the results of the classification. As is known, HRV and PRV parameters have values that depend on individual factors, such as sex, age, time of day, and physical activity. First, we made a stratification of the data according to the sex of our volunteers. The male–female distribution of some PRV features is shown in Figure 13. Our trends are like those described in [27].

In Figure 14, there are some more details about the classification results. The 238 observations were divided into two groups: 109 belong to men, 129 to women. The confusion matrices show the results of classification using: the KNN algorithm, PRV + SP features, and LOSO cross-validation. There is only one statistically significant difference between the results for men and women: the classification of pleasantness (“unpleasant” class).

In Figure 15 the 238 observations were divided into two groups: 125 belong to anxious volunteers (State anxiety > 45), and 113 belong to calm volunteers (State anxiety < 45). The confusion matrices show the results of classification using: the KNN algorithm, PRV + SP features, and LOSO cross-validation. There is only one statistically significant difference between the results for anxious and calm subjects: the classification of dominance (“in control” class). 

We analyzed the data in more detail to understand the extent to which the state of anxiety can determine the accuracy of the classification. In Figure 16, there are the correlation plots of accuracy and anxiety values, referred to as the dominance emotional variable. The accuracy of the classification is significantly reduced as state anxiety increases, especially if the SVM algorithm is used. The KNN algorithm seems less sensitive to individual disturbing factors.

## 4. Discussion

Differences in HRV and PRV have been studied under various physiological conditions. Mejia-Mejia and colleagues [28] investigated the differences between HRV and PRV, in a whole-body cold exposure (CE) experiment. Most of the PRV time-domain and Poincaré plot indices increased during cold exposure. HRV-derived parameters showed a similar behavior but were less affected than PRV. Frequency-domain absolute power indices (i.e., LF, HF, and total power TP) showed the worst agreement between HRV and PRV, as in our experiment. However, they found that PRV generally overestimates HRV indices, especially under cold exposure.

The HRV and the PRV measures obtained from the index fingers of both hands were compared in the work of Wong and colleagues [31]. The values of LF and HF obtained from ECG and PPG exhibit significant differences. Additionally, insufficient agreement was found in the pairwise comparisons of left PRV versus right PRV. The authors of that work concluded that, for this reason, PRV cannot be used as a surrogate for HRV.

We experimentally found a good match between the HRV parameters and the corresponding PRV parameters, with some exceptions as reported in the previous section (Figure 7, Figure 8 and Figure 9). These differences have already been reported in the literature in volunteers subjected to physiological stimuli of a different nature (not emotional) [26,40]. The use of PPG as an ECG surrogate must be evaluated case by case. It is more convenient to think of the PPG signal as a different biomarker, which provides a richer set of information than what can be derived from the IBI sequence. [42]

We have shown that PPG technology can be used for emotion recognition instead of ECG technology with comparable, slightly better, results. The use of shape parameters derived from the PPG signal, combined with the more traditional HRV parameters, derived from beat duration, is advantageous. It, therefore, appears that the form of the PPG signal contains additional physiological information that is useful for emotion recognition.

In fact, the shape of the PPG signal and its variation are determined by the electrical input that arrives at the sino-atrial node of the heart, but also by inputs that arrive directly to the arteries, and many other factors, like arterial stiffness. What is interesting is that the simultaneous presence of so many concomitant factors is not a source of confusion, but rather contributes to improving the recognition of emotions.

The results we obtained are in line with the most recent literature. Here, we cite some examples of studies where a procedure comparable to our work was used. 

In the article by Sepulveda and colleagues [43] the physiological signals of a public dataset AMIGOS [44] were used. Valence and Excitation binary classes were evaluated with a parameter extraction procedure using wavelet transform scattering. For comparison, the classification with traditional HRV parameters was also made. Six of the most common classification algorithms were tested. 

Using HRV parameters, accuracy values were obtained in the range of 52–59% for valence and values in the range of 57–60% for arousal. 

Wei and colleagues [45], using the MANHOB-HCI [46] dataset, developed an emotion recognition model based on the fusion of multichannel physiological signals. In their study, they attempted to classify five emotions, according to the Discrete Emotional Model by Ekman [47]: Sadness, Happiness, Disgust, Neutral, and Fear. Based only on the ECG signals, they obtained an average recognition rate (accuracy) of 68.75%. With their four-signal fusion model (EEG, ECG, respiration, and GSR), the accuracy reaches 84.62%.

Ferdinando and colleagues [48], using ECG signals from the MANHOB-HCI [44] database and features from standard Heart Rate Variability analysis, obtained average accuracies for valence and arousal of 42.6% and 47.7%, respectively, with 10-fold cross-validation. Much better accuracy values were obtained using methods other than the traditional HRV analysis. Selvaraj and colleagues [49] induced six basic emotional states (happiness, sadness, fear, disgust, surprise, and neutral) using audio-visual stimuli. The non-linear feature ‘Hurst’ and Higher Order Statistics (HOS) were computed. The features were then classified using several classifiers. An accuracy of 92.87% for classifying the six emotional states was obtained using a Fuzzy KNN classifier with random cross-validation. The accuracy is reduced to 76.45% using a subject-independent validation (Leave-one-subject-out, LOSO). 

Valenza and colleagues [24] proposed a personalized probabilistic framework able to characterize the emotional state of a subject through the analysis of heartbeat dynamics exclusively. They achieved an average accuracy of 71.43% in recognizing self-reported emotions with a non-linear feature set. The accuracy is reduced to 67.19% using only linear features. LOSO cross-validation was used.

This brief review of emotion classification studies, based on ECG only, makes no claim to completeness. We undertook it with the sole purpose of verifying whether the accuracy values we obtained are in line with what is reported in the literature.

## 5. Conclusions

The traditional parameters derived from ECG/PPG signals in the time domain and in the frequency domain (SDNN, RMSSD, LH, HF, …) are still considered the most obvious and natural choice to describe HRV. For this reason, using these traditional parameters for an ECG versus PPG comparison makes sense. 

This study examined whether the use of PPG sensors can be effective in detecting emotional states, such as anger, overexcitement, and anxiety. For this purpose, a range of emotional responses was elicited in thirty-one young, healthy subjects.

The detection rate of the main emotional variables obtained with the PPG technology was compared and statistically analyzed, with the detection rate obtained using ECG technology. It is confirmed that with PPG technology, we can obtain the recognition rates of emotional variables comparable to those obtainable with the more traditional ECG technology. We made an equal comparison by extracting from ECG and PPG signals the parameters that are commonly referred to as HRV and PRV parameters, and we found that they have similar values and trends (with some exceptions), as well as similar effectiveness in discriminating the levels of some emotional variables. 

There are large disturbances in HRV measurements, due to individual differences, as can be easily seen in Figure 6. Individual differences can be compensated to a certain extent by the normalization procedure we adopted. Our experimental procedure highlighted an important disturbing factor: the individual’s state of anxiety and mood, which we have measured with psychological tests (POMS and STAI -Y). The effect of this factor can be seen in the Figure 14, Figure 15, and Figure 16. The ability (of the algorithms we use) to predict the emotional response degrades as the state of anxiety increases. This is clear evidence that physiological parameters are driven both by the mood of the individuals and by the stimuli. Future studies on emotion recognition should carefully take into account this factor and introduce the measure of state anxiety as a standard step of the experimental procedure.

The main finding of our work is the following: we have shown that from the PPG signal, it is possible to extract new parameters related to the shape of the pulse, not derived from the period, like the traditional HRV parameters. These new shape parameters (SP) have been found to be effective in recognizing levels of emotional variables. 

The variability of the parameters obtained from PPG, both the traditional PRV and those derived from the shape, is controlled by the inputs of the autonomic nervous system, such as HRV. However, the ways through which this control is carried out are different. The electrical signal comes directly from the heart and indicates the interaction of the autonomic nervous system with the heart’s nerve centers. The optical signal derives from the oscillations of the heart but also from the arteries, which have nerve endings independent from those of the heart, and which receive input from the autonomic nervous system through different nerve endings. 

We have shown that it is possible to measure the response of the autonomic nervous system to emotional stimuli thanks to a physiological effect, so far not considered or little considered (to our knowledge). 

The circulation of the blood in the peripheral vessels is favored/disadvantaged by a low/high stiffness of the arteries. Our study indicates that the stiffness of the arteries, mainly linked to factors such as age, atherosclerosis, etc., is also modulated by responses to emotional stimuli. This modulation effect is new, or at least rarely reported in the literature. The autonomic nervous system, in response to emotional stimuli, acts in two ways: it changes the rhythm of the heart (HR and HRV) and, at the same time, changes the stiffness of the arteries. This change is temporary and reversible. The stiffness of the arteries is easily detectable because it modifies the shape of the PPG pulse. Thus, the shape parameters (SP) are a different biomarker from HRV and PRV but are useful for revealing emotional responses.

Even if these results were obtained from a homogeneous sample of young adults, they are promising in view of further validations with other age cohorts and people with pathological health conditions. 

Ultimately, this work represents the first step of a larger project, which aims to diversify the sample and use a multimodal system that includes other psychophysiological detection techniques, such as EEG and GSR.

## Figures and Tables

**Figure 1 biosensors-12-00811-f001:**
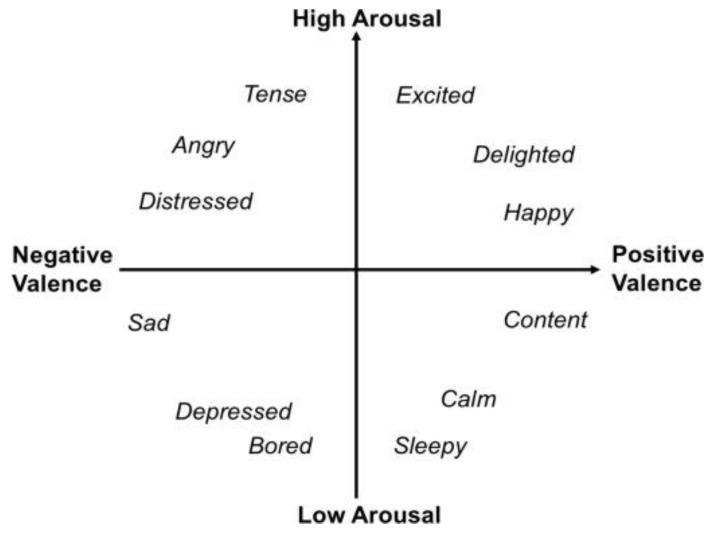
Russell’s circumplex model of affect [20].

**Figure 2 biosensors-12-00811-f002:**
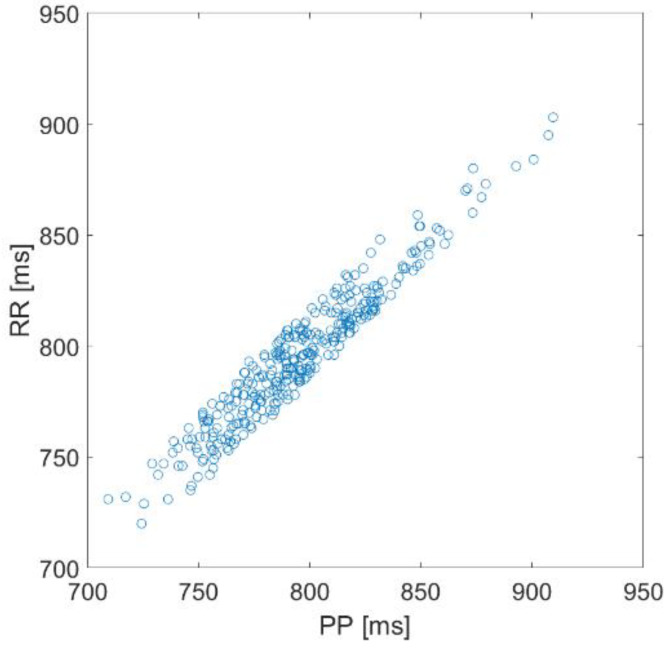
IBI values obtained by a synchronous acquisition of ECG and PPG signals; two minutes acquisition. PP: IBI value derived from PPG; RR: IBI value derived from ECG.

**Figure 3 biosensors-12-00811-f003:**
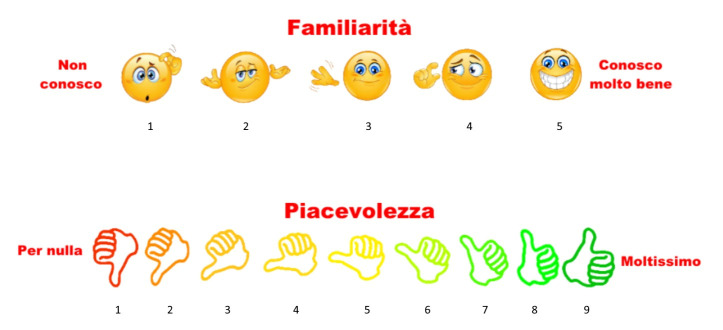
Self-Assessment Manikin (Familiarity and pleasantness). Screenshot of the user interface.

**Figure 4 biosensors-12-00811-f004:**
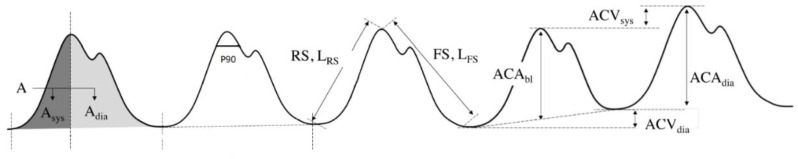
Graphical representation of the parameters extracted with a morphological analysis of the PPG signal. Adapted from: [40].

**Figure 5 biosensors-12-00811-f005:**
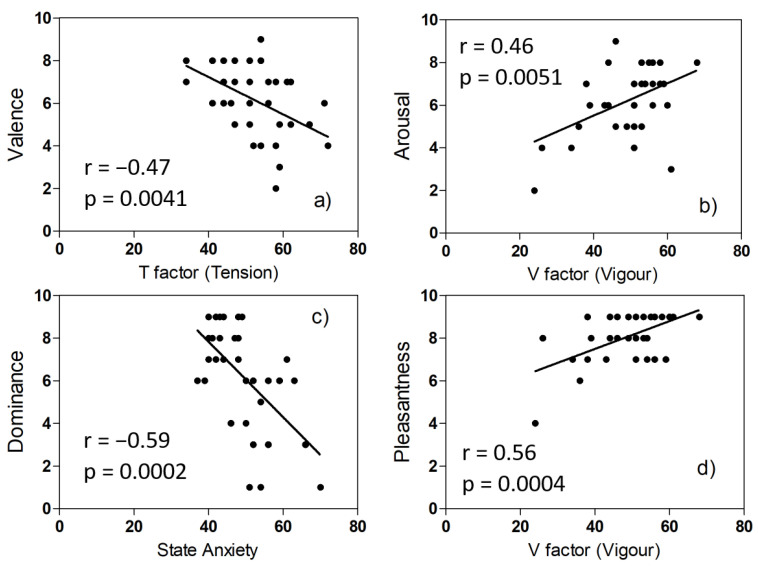
Correlation plots of SAM scores and mood or anxiety values: (**a**) video F (low arousal, high valence), (**b**) video D (high arousal, high valence), (**c**) video A (low arousal, low valence), (**d**) video D (high arousal, high valence).

**Figure 6 biosensors-12-00811-f006:**
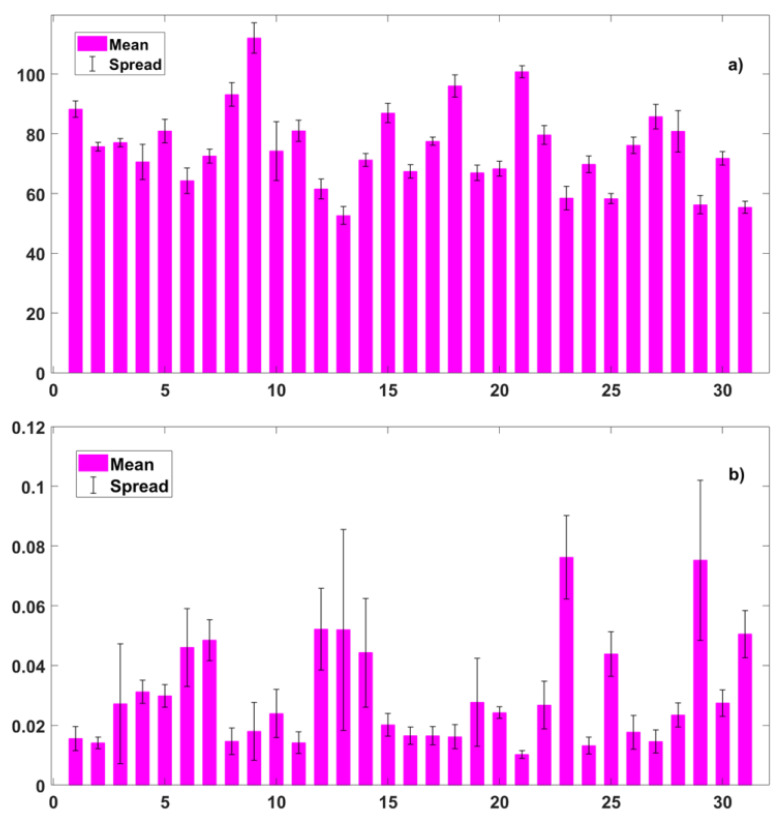
Examples of intra- and inter-subject parameter changes. For each subject, the mean value (red bars) and spread (black lines) are shown. Features: (**a**) HR PPG, (**b**) SD1 PPG.

**Figure 7 biosensors-12-00811-f007:**
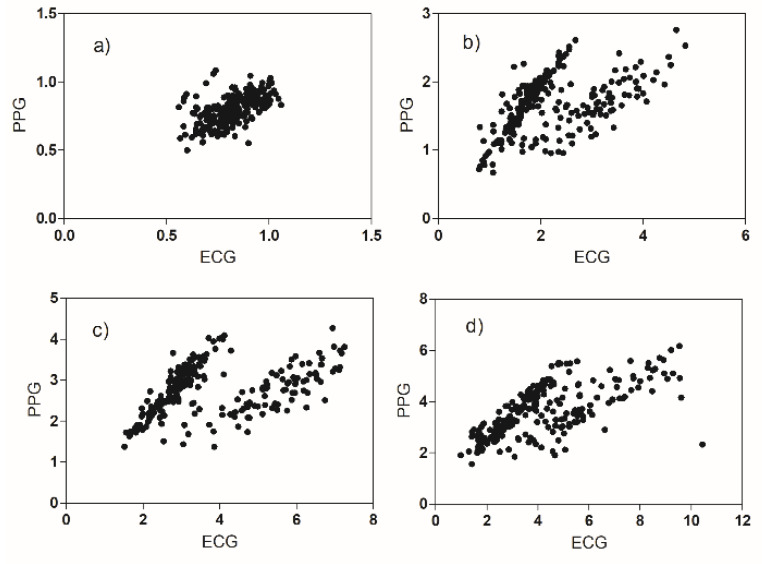
Scatter plots for (**a**) ApEn, (**b**) VLF, (**c**) LF, (**d**) HF values derived from ECG and PPG sensors.

**Figure 8 biosensors-12-00811-f008:**
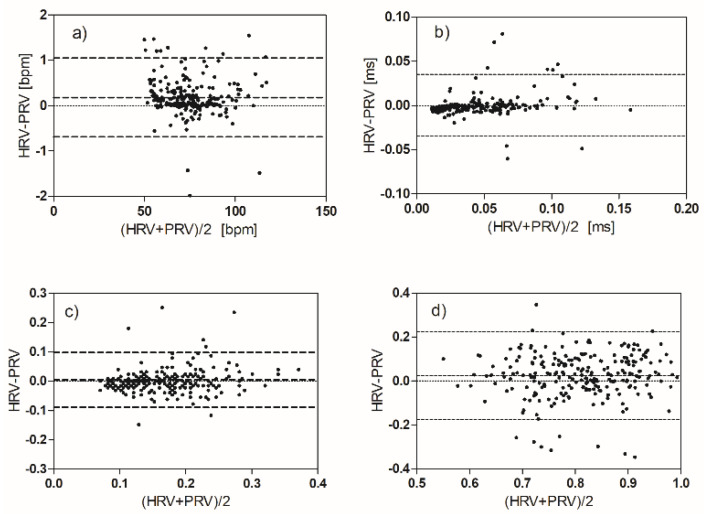
Bland−Altman plots comparing HRV and PRV measurements: (**a**) HR, (**b**) RMSSD, (**c**) TINN, (**d**) ApEn.

**Figure 9 biosensors-12-00811-f009:**
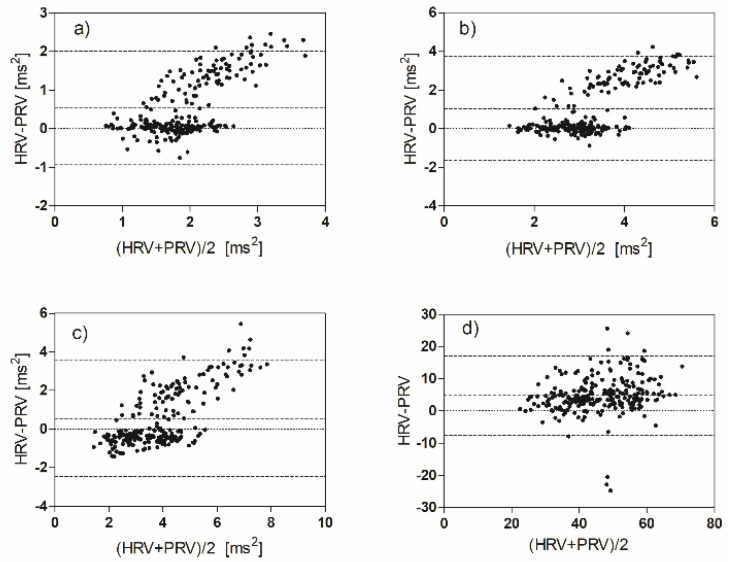
Bland−Altman plots comparing HRV and PRV measurements: (**a**) VLF, (**b**) LF, (**c**) HF, (**d**) pLF.

**Figure 10 biosensors-12-00811-f010:**
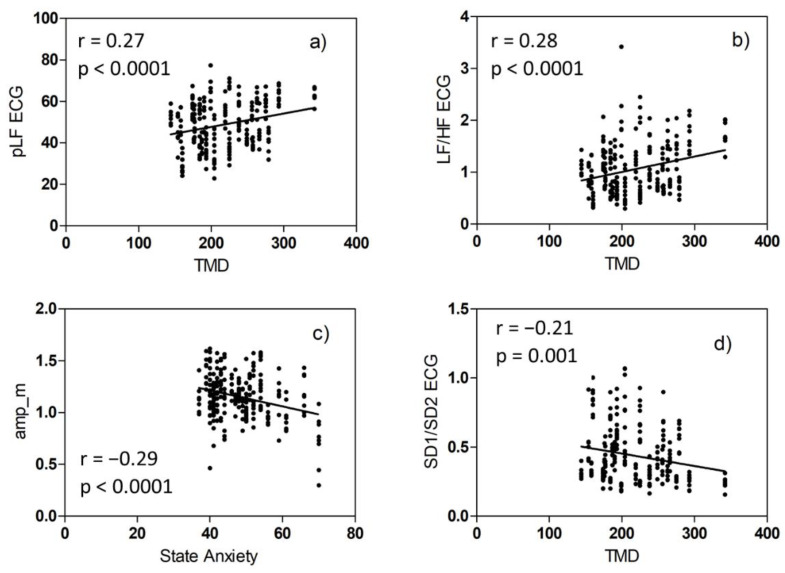
Correlation plots of HRV or SP values and TMD or STAI−Y values: (**a**) pLF ECG, (**b**) LF/HF ECG, (**c**) amp_m, (**d**) SD1/SD2 ECG.

**Figure 11 biosensors-12-00811-f011:**
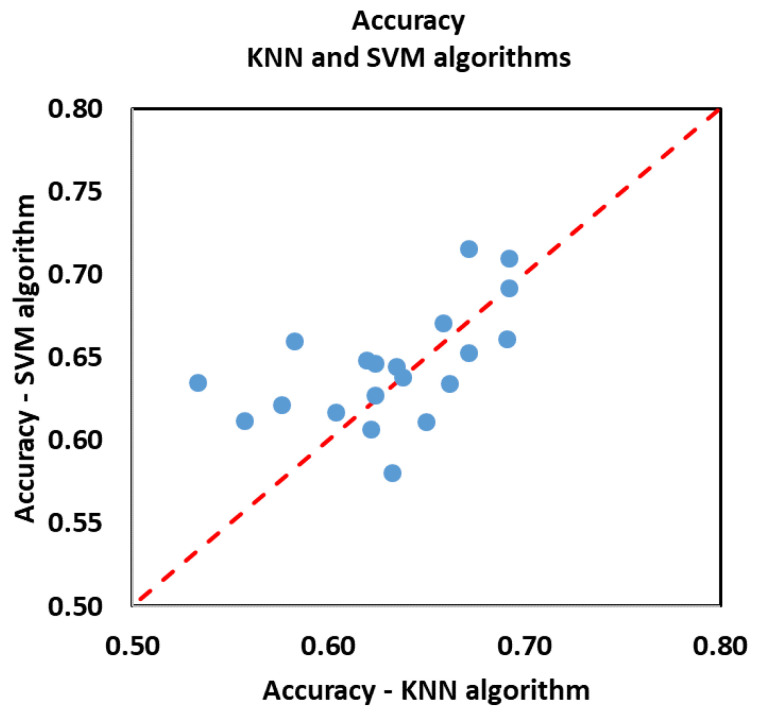
Accuracy values. Each point represents the accuracy with which one of the five emotional variables was classified, having used one of the four sets of features.

**Figure 12 biosensors-12-00811-f012:**
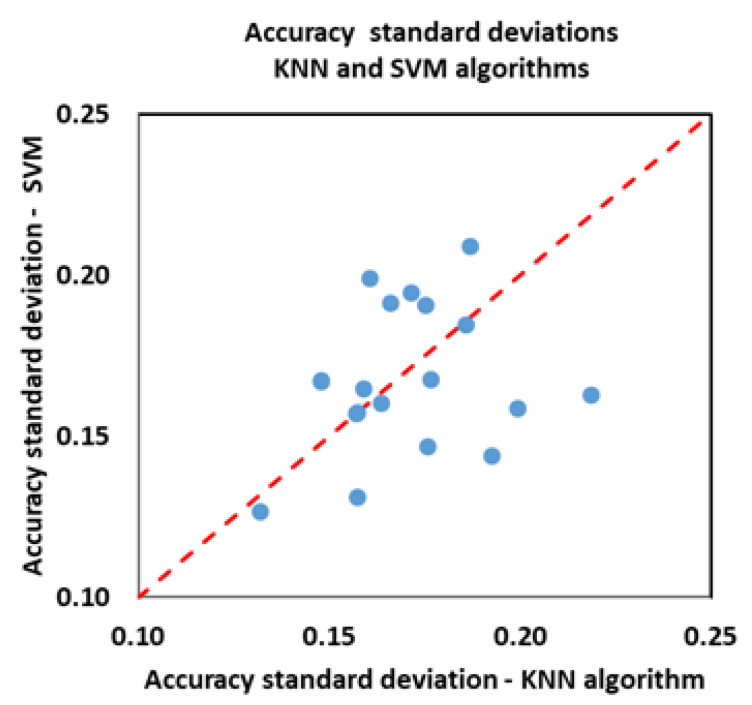
Accuracy standard deviation. Each point represents the accuracy standard deviation with which one of the five emotional variables was classified, having used one of the four sets of features.

**Figure 13 biosensors-12-00811-f013:**
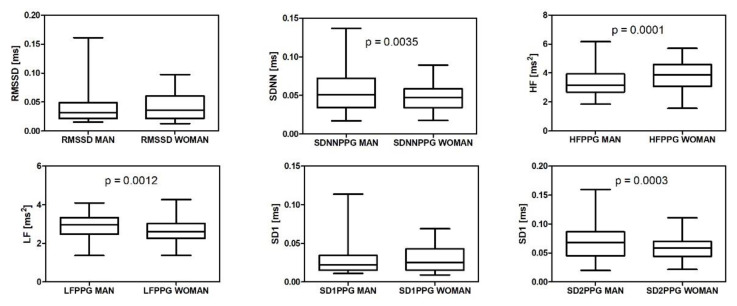
Analysis of six PRV features. Values distributed into the two classes: Man, Woman. Unpaired *t*-test, two-tailed. The means are significantly different in four cases: SDNN, HF, LF, and SD2.

**Figure 14 biosensors-12-00811-f014:**
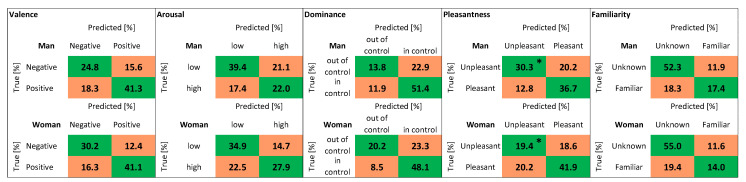
Confusion matrices. Values distributed into the two classes: Man, Woman. Classification obtained with: KNN algorithm, PRV + SP features, LOSO cross-validation. The values marked with one asterisk, in the box relating to the Pleasantness variable, are significantly different (Mann–Whitney test, two-tailed, based on the values that refer to each volunteer). *p* = 0.038.

**Figure 15 biosensors-12-00811-f015:**
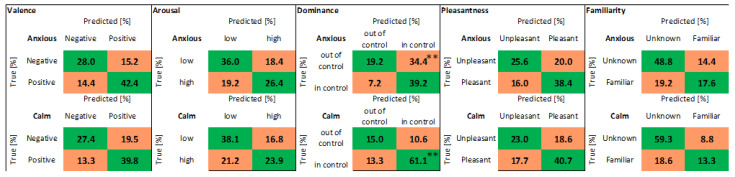
Confusion matrices. Values distributed into the two classes: Anxious, Calm. Classification obtained with: KNN algorithm, PRV + SP features, LOSO cross-validation. The values marked with two asterisks, in the box relating to the dominance variable, are significantly different (Mann–Whitney test, two-tailed, based on the values that refer to each volunteer). *p* = 0.0064.

**Figure 16 biosensors-12-00811-f016:**
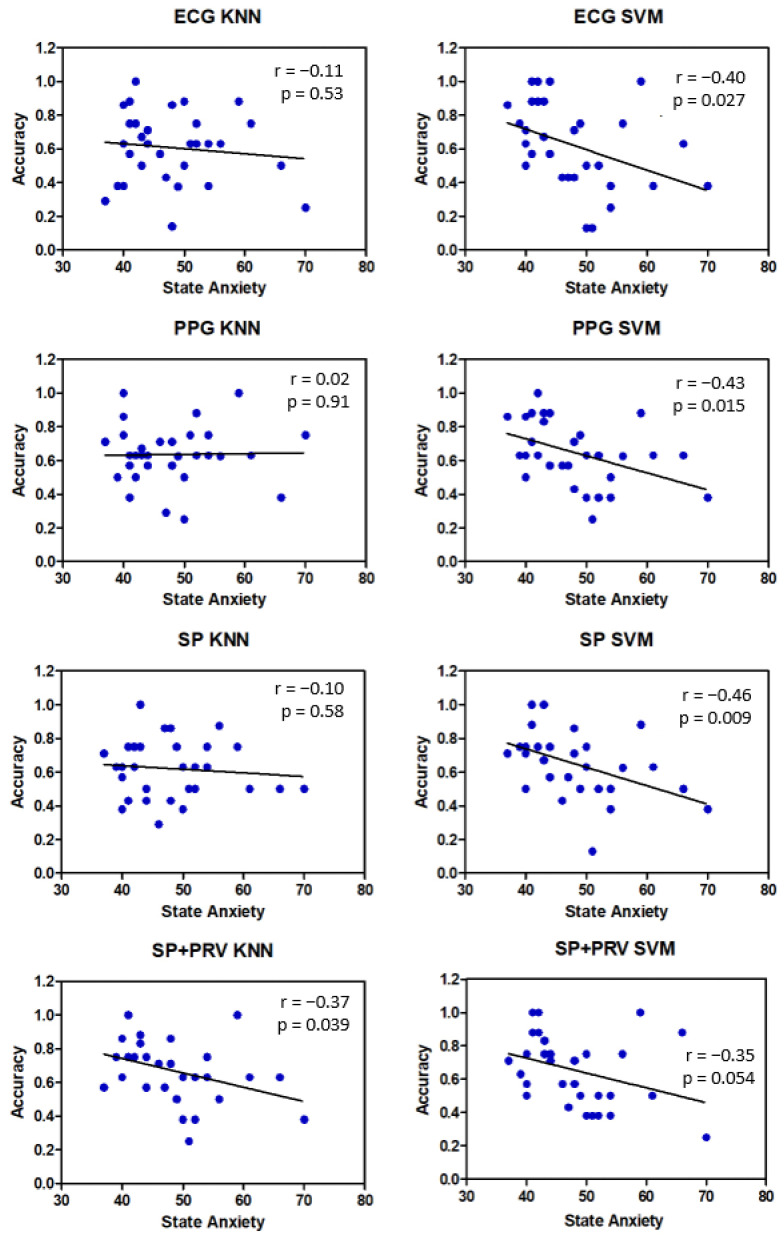
Classification of Dominance emotional variable. Correlation plots of accuracy and anxiety values. The 31 accuracy values refer to individual volunteers.

**Table 1 biosensors-12-00811-t001:** HRV indices based on IBI intervals.

Parameter	Unit	Description
SDNN	ms	Standard deviation of NN intervals ^a^
RMSSD	ms	Root mean square of successive RR intervals differences
TRI		Integral of the density of the RR interval histogram divided by its height
TINN	ms	Baseline width of the RR interval histogram
ApEn		Approximate entropy, which measures the regularity and complexity of a time series
SD1	ms	Poincaré plot standard deviation perpendicular to the line of identity
SD2	ms	Poincaré plot standard deviation along the line of identity
SD1/SD2		Ratio of SD1-to-SD2
HR	bpm	Heart Rate
VLF	ms^2^	Power in VLF range (<0.04 Hz)
LF	ms^2^	Power in LF range (0.04–0.15 Hz)
HF	ms^2^	Power in HF range (0.15–0.4 Hz)
pLF		LF power in normalized units
pHF		HF power in normalized units
LF/HF		Ratio LF (ms^2^)/HF (ms^2^)

^a^ NN intervals: interbeat intervals from which artifacts have been removed.

**Table 2 biosensors-12-00811-t002:** Shape parameters.

Parameter	Definition	Description
Asys	Area of a systolic phase	The area of a pulse from the diastolic peak to the next systolic peak
Adia	Area of a diastolic phase	The area of a pulse from the systolic peak to the next diastolic peak
ACAbl	AC Amplitude from baseline	Difference of the systolic peak amplitude and the interpolated baseline amplitude of two adjacent diastolic peaks
ACVsys	AC Variation Systole	Difference of the amplitude of systolic peaks
ACVdia	AC Variation Diastole	Difference of the amplitude of diastolic peaks
Lrs	Rising Slope Length	Distance between the diastolic peak and the next systolic peak
Lfs	Falling Slope Length	Distance between the systolic peak and the next diastolic peak
ACAdia	AC Amplitude from the previous diastole	Difference of the systolic peak amplitude and the previous diastolic peak amplitude
RS	Rising Slope	Slope between the diastolic peak and the next systolic peak
P90	Pulse Width at 90% of amplitude	Pulse width at 90% point of maximum amplitude
R1	ACVsys/ACAdia	Ratio of AC variation systole to head peak height
R2	ACVsys/ACAbl	Ratio of AC variation systole to AC amplitude
R3	ACVdia/ACAdia	Ratio of AC variation diastole to head peak height
R4	ACVdia/ACAbl	Ratio of AC variation diastole to AC amplitude
AsAd	Asys/Adia	Ratio of systolic area to diastolic area
LrLs	Lrs/Lfs	Ratio of rising slope length to falling slope length

**Table 3 biosensors-12-00811-t003:** Binary classes.

Emotional Variables	SAM Scores	Classes	Observations	[%]
Valence	1 ÷ 5	Negative	104	43.7
	6 ÷ 9	Positive	134	56.3
Arousal	1 ÷ 5	Low	142	59.7
	6 ÷ 9	High	96	40.3
Dominance	1 ÷ 5	Out of control	76	31.9
	6 ÷ 9	In control	162	68.1
Pleasantness	1 ÷ 5	Unpleasant	105	44.1
	6 ÷ 9	Pleasant	133	55.9
Familiarity	1 ÷ 2	Unknown	171	71.8
	3 ÷ 5	Familiar	67	28.2

**Table 4 biosensors-12-00811-t004:** Linear regression of HRV features versus PRV features.

Feature	Best Fit Equation	r (Pearson)	*p* Value
SDNN	y = 0.01 + 0.78x	0.91	<0.0001
RMSSD	y = 0.01 + 0.69x	0.82	<0.0001
TRI	y = 2.75 + 0.74x	0.86	<0.0001
TINN	y = 0.05 + 0.69x	0.77	<0.0001
ApEn	y = 0.36 + 0.53x	0.55	<0.0001
SD1	y = 0.01 + 0.69x	0.82	<0.0001
SD2	y = 0.01 + 0.79x	0.94	<0.0001
SD1/SD2	y = 0.11 + 0.83x	0.84	<0.0001
HR	y = −0.23 + x	0.99	<0.0001
pLF	y = 8 + 0.73x	0.83	<0.0001
pHF	y = 18.2 + 0.74x	0.83	<0.0001
LF/HF	y = 0.26 + 0.55x	0.78	<0.0001
VLF	y = 1.13 + 0.24x	0.51	<0.0001
LF	y = 2.16 + 0.16x	0.40	<0.0001
HF	y = 2.24 + 0.33x	0.67	<0.0001

**Table 5 biosensors-12-00811-t005:** One-way ANOVA analysis of binary classes.

Features	Valence		Arousal		Dominance		Pleasantness		Familiarity	
	ECG	PPG	ECG	PPG	ECG	PPG	ECG	PPG	ECG	PPG
RMSSD	0.019	0.045			0.022	0.006				
SD1	0.019	0.045								
HR	0.003	0.002	0.022	0.024						
HF				0.047					0.003	0.002
TINN					0.023					
TRI					0.029					
SD2								0.021		
pLF									0.030	0.021
pHF									0.030	0.021
LF/HF									0.021	0.021

Only significant *p*-values are shown.

**Table 6 biosensors-12-00811-t006:** Classification metrics: HRV versus PRV features.

**Weighted kNN**		**HRV**				**PRV**			
		**Precision**	**Recall**	**F1**	**Accuracy**	**Precision**	**Recall**	**F1**	**Accuracy**
Valence	Positive	0.59	0.65	0.62	0.56 (0.22)	0.60	0.69	0.64	0.58 (0.16)
	Negative	0.51	0.45	0.48		0.53	0.43	0.47	
Arousal	High	0.59	0.61	0.60	0.63 (0.17)	0.59	0.56	0.57	0.62 (0.15)
	Low	0.67	0.65	0.66		0.65	0.68	0.66	
Dominance	In control	0.67	0.65	0.66	0.61 (0.21)	0.68	0.74	0.71	0.63 (0.18)
	Out of control	0.51	0.53	0.52		0.55	0.48	0.51	
Pleasantness	Pleasant	0.63	0.65	0.64	0.58 (0.18)	0.63	0.64	0.63	0.58 (0.19)
	Unpleasant	0.53	0.50	0.51		0.52	0.51	0.51	
Familiarity	Familiar	0.51	0.34	0.41	0.66 (0.13)	0.53	0.41	0.46	0.67 (0.16)
	Unknown	0.70	0.83	0.76		0.72	0.81	0.76	
Gaussian SVM		HRV				PRV			
		Precision	Recall	F1	Accuracy	Precision	Recall	F1	Accuracy
Valence	Positive	0.63	0.73	0.68	0.61 (0.16)	0.65	0.66	0.65	0.62 (0.20)
	Negative	0.59	0.48	0.53		0.58	0.56	0.57	
Arousal	High	0.44	0.55	0.49	0.58 (0.19)	0.63	0.56	0.59	0.65 (0.17)
	Low	0.70	0.60	0.65		0.66	0.72	0.69	
Dominance	In control	0.61	0.97	0.75	0.62 (0.25)	0.65	0.89	0.75	0.64 (0.19)
	Out of control	0.69	0.09	0.16		0.63	0.28	0.39	
Pleasantness	Pleasant	0.63	0.83	0.72	0.63 (0.17)	0.67	0.77	0.72	0.66 (0.14)
	Unpleasant	0.63	0.38	0.47		0.64	0.52	0.57	
Familiarity	Familiar	0.59	0.16	0.25	0.67 (0.13)	0.70	0.32	0.44	0.72 (0.16)
	Unknown	0.68	0.94	0.79		0.72	0.93	0.81	
Standard deviations of the accuracy are inside brackets									

**Table 7 biosensors-12-00811-t007:** Classification metrics: shape parameters (SP) versus PRV + SP.

**Weighted kNN**		**SP**				**PRV + SP**			
		**Precision**	**Recall**	**F1**	**Accuracy**	**Precision**	**Recall**	**F1**	**Accuracy**
Valence	Positive	0.68	0.73	0.70	0.66 (0.18)	0.71	0.74	0.72	0.69 (0.16)
	Negative	0.64	0.58	0.61		0.67	0.62	0.64	
Arousal	High	0.58	0.58	0.58	0.62 (0.19)	0.59	0.56	0.57	0.62 (0.15)
	Low	0.65	0.65	0.65		0.65	0.68	0.66	
Dominance	In control	0.68	0.69	0.68	0.62 (0.17)	0.68	0.83	0.75	0.67 (0.19)
	Out of control	0.53	0.51	0.52		0.63	0.43	0.51	
Pleasantness	Pleasant	0.68	0.75	0.71	0.66 (0.20)	0.67	0.70	0.68	0.64 (0.16)
	Unpleasant	0.62	0.54	0.58		0.59	0.56	0.57	
Familiarity	Familiar	0.57	0.45	0.50	0.69 (0.16)	0.57	0.41	0.45	0.69 (0.16)
	Unknown	0.74	0.82	0.78		0.74	0.79	0.82	
Gaussian SVM		SP				PRV + SP			
		Precision	Recall	F1	Accuracy	Precision	Recall	F1	Accuracy
Valence	Positive	0.64	0.77	0.70	0.63 (0.15)	0.65	0.82	0.73	0.66 (0.16)
	Negative	0.63	0.47	0.54		0.68	0.47	0.56	
Arousal	High	0.59	0.44	0.50	0.61 (0.18)	0.62	0.48	0.54	0.63 (0.17)
	Low	0.62	0.75	0.68		0.64	0.75	0.69	
Dominance	In control	0.65	0.87	0.74	0.65 (0.19)	0.65	0.90	0.90	0.65 (0.21)
	Out of control	0.63	0.31	0.42		0.66	0.28	0.39	
Pleasantness	Pleasant	0.62	0.78	0.69	0.61 (0.16)	0.67	0.77	0.72	0.66 (0.16)
	Unpleasant	0.58	0.39	0.47		0.64	0.52	0.57	
Familiarity	Familiar	0.84	0.78	0.20	0.71 (0.13)	0.70	0.32	0.44	0.72 (0.16)
	Unknown	0.70	0.39	0.98		0.72	0.93	0.81	
Standard deviations of the accuracy are inside brackets									

**Table 8 biosensors-12-00811-t008:** Relevant features.

**Weighted kNN**	**HRV**	**PRV**	**SP**	**PRV + SP**
Valence	SD1SD2R, VLF	HR, SDNN, LFHFratio	RMSSD_AsAd	HR, RMSSD, RMSSD_R3, RMSSD_R4
Arousal	SDNN, SD2, VLF, LF	SDNN, SD1	R1_m	SDNN, SD1
Dominance	LFHFratio	RMSSD, pLF	R2_m, amp_m	RMSSD, pLF, RMSSD_R2, RMSSD_LrLs
Pleasantness	HF	LF	RMSSD_AsAd, RMSSD_R3, LrLs_m	LF, LFHFratio, AsAd_m
Familiarity	ApEn, VLF, HF	RMSSD, SD1, SD2	RMSSD_R3, LrLs_m	RMSSD_R3, LrLs_m
Gaussian SVM	HRV	PRV	SP	PRV + SP
Valence	HR, TRI, SD2, LFHFratio, VLF, LF	TRI, ApEn	RMSSD_LrLs, RMSSD_amp, P90_m	pLF, pHF, P90_m
Arousal	HR, SDNN, SD1SD2R, LF, VLF	TINN, SD2, LF	RMSSD_AsAd	SDNN, RMSSD_AsAd, RMSSD_amp
Dominance	RMSSD	SDNN, RMSSD	RMSSD_AsAd, RMSSD_RS	RMSSD, RMSSD_RS, R3_m
Pleasantness	SDNN, TINN, apen, SD2, SD1SD2R, pLF, LFHFratio, VLF, LF	SD1, SD2	RMSSD_LrLs, P90_m	SD1, SD2
Familiarity	SD1SD2R, LF	RMSSD, SD1, SD2	RMSSD_R4, LrLs_m	RMSSD, SD1, SD2

## Data Availability

The raw data are available on request. Please, for any requests, contact P.G.F.

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
