# Peer review of "Emotion Recognition: Photoplethysmography and Electrocardiography in Comparison"

_biosensors, 2022, doi:10.3390/bios12100811_

Round 1
Reviewer 1 Report
The manuscript is interesting and reinforces our knowledge about emotion recognition and the comparison of two different techniques: photoplethysmography and electrocardiography. The authors provide an introduction that elaborates on previously published literature and justifies their interest in conducting the study. The sequence of data collection is described in detail in the material and methods section. Data presentation in the results section is very comprehensive and well structured. However, there are some aspects that should be modified:
- Material and methods: it is not very clear why a sample of 40 volunteers is used first and then another 31 subjects.
- Results section: In table 1: authors should define some parameters such as LF, VLF,..
- Results section: the authors discuss results derived from other authors' studies. I consider that these comments should be part of the discussion section.
- Discussion section: authors should add others possible limitations of the study.

Author Response
Point 1: Material and methods: it is not very clear why a sample of 40 volunteers is used first and then another 31 subjects.
Response 1: Thank you for the comment that allows a better understanding of the procedure used. In section 2.1 Emotion elicitation protocol, the procedures for selecting the final video clips and the two samples used have been better clarified in the text. In this regard, in accordance with the work of Kolestra et al., 2006 (see Koelstra, S., et al. Deap: A database for emotion analysis; using physiological signals. IEEE Trans. Affect. Comput., vol. 3, n. 1, pp. 18-31, 2011) on which we relied, the sample of 40 subjects was used simply to receive a preliminary evaluation of the pre-selected 40 video clips that would allow us to select the best performing ones, the final 8 used for the experimental session here. So, the sample of 40 subjects is independent of the sample of 31 subjects recruited later and who participated in the study presented here.
Point 2: Results section: In table 1: authors should define some parameters such as LF, VLF,..
Response 2: I agree with your comment. The definitions are incomplete. In the revised text we will add a few sentences to better define the parameters used. Regarding the parameters in the time domain, a concise definition is the following.
For frequency-domain analysis, traces were interpolated using cubic-spline interpolation, and the power spectra were obtained using fast Fourier transform (FFT). The absolute and relative powers of very low frequency (VLF; < 0.04 Hz); low-frequency (LF; 0.04–0.14 Hz) and high-frequency (HF; 0.15–0.4 Hz) bands were measured.
Point 3: Results section: the authors discuss results derived from other authors' studies. I consider that these comments should be part of the discussion section.
Response 3: I agree with your comment. I will move lines 497-509 of page 15 to the discussion section.
Point 4: Discussion section: authors should add others possible limitations of the study.
Response 4: Several limitations of this study should be highlighted. The experimental procedure we have adopted takes a long time for each participant. So, we were able to study volunteers of only one age group. The number of volunteers we recruited was not small in comparison with similar studies, but after analysing the results, we realized that a greater number of subjects is needed to also consider the general conditions of stress. The number of participants was enough to highlight this effect, but experiments with a higher number will be needed to study it thoroughly.
Next, during the presentation of the film clips the subjects sit still and the amount of movements was small. Under these conditions the parameters PRV and HRV give similar information. A field study, which will certainly be more affected by disturbances in the signal, due to motion artifacts, could give different indications.
Future work will include experimenting with instrumentation to determine such physiological signals as GSR, body temperature and respiration rate. The results may be combined with this experiment to raise the overall accuracy level.
Reviewer 2 Report
l This paper uses the most classical linear HRV / PRV measurement index and SVM / KNN classical classifier. In fact, the nonlinear HRV measurement method has attracted wide attention and has been proved to be superior in overcoming various interferences. E.g. Classifification of Sleep Apnea Severity by Electrocardiogram Monitoring Using a Novel Wearable Device, Sensors 2020. ...Teporal dependecy complexity: The Novel Approach of Temporal Dependency Complexity Analysis of Heart Rate Variability in Obstructive Sleep Apnea, Computers in biology and medicine 2021...
l Deep leaning methods were widely used in ECG/PPG classifer. E.g. Obstructive sleep apnea detection from single-lead electrocardiogram signals using one-dimensional squeeze-and-excitation residual group network. Computers in biology and medicine 2022(CNN).....
l
1) The method motivation of the paper is missing. It is suggested that the reason for method selection should be explained in the discussion, and the nonlinear HRV measurement and deep learning algorithm should be compared or analyzed to extract the contribution of the paper
2) There are large disturbances in HRV measurements, such as individual differences, age / sex. E.g. Robustness Evaluation of Heart Rate Variability Measures for age and gender related autonomic changes in Healthy Volunteers. How to suppress these influencing factors, or what is the difference of HRV measurement between these factors and emotion.
3) The main contribution of this paper is not clear. It is suggested to refine the innovation of this paper more clearly through peer comparison (including HRV measurement methods, classification methods, new mechanism discoveries, etc.)
4) What are the advantages and disadvantages of PRV measurement compared to HRV. Can we fuse the two signals to improve the performance of emotion recognition?
5) It is suggested to introduce the data in detail in the form of a table, draw a confusion matrix for the classification results, and conduct cross validation to improve the reliability of the classifier
Author Response
Response to Reviewer 3 Comments
Point 1: The method motivation of the paper is missing. It is suggested that the reason for method selection should be explained in the discussion, and the nonlinear HRV measurement and deep learning algorithm should be compared or analyzed to extract the contribution of the paper
Response 1: Thank you for the comment: indeed, the choice of the classification method and its motivation are not sufficiently explained in the text. Our aim was to use a straightforward algorithm to compare parameters extracted from the ECG and PPG waveforms acquisitions. So, we used the most traditional and most popular PRV and HRV parameters. A comparison between several algorithms was not the primary focus of our work. Then we used the most popular algorithms in the field of emotion recognition: SVM and KNN.
Point 2: There are large disturbances in HRV measurements, such as individual differences, age / sex. E.g. Robustness Evaluation of Heart Rate Variability Measures for age and gender related autonomic changes in Healthy Volunteers. How to suppress these influencing factors, or what is the difference of HRV measurement between these factors and emotion.
Response 2: Our data does not show a different response to emotional stimuli due to sex. We cannot say anything about the weight of the age factor, because we only tested a homogeneous group of young adults. Certainly, there are strong individual differences, as can be easily seen in figure 6. Furthermore, there is an important disturbing factor: the individual's state of anxiety and mood, which we have measured with psychological tests (POMS and STAI -Y). The effect of this factor can be seen in figure 11. The ability (of the algorithms we use) to predict the emotional response degrades as the state of anxiety increases. Individual differences can be compensated to a certain extent by the normalization procedure, as we briefly described in lines 364-367 on page 10.
Point 3: The main contribution of this paper is not clear. It is suggested to refine the innovation of this paper more clearly through peer comparison (including HRV measurement methods, classification methods, new mechanism discoveries, etc.)
Response 3: The main contribution of our article is the following. We have shown that it is possible to measure the response of the autonomic nervous system to emotional stimuli thanks to a physiological effect, so far not considered or little considered (to our knowledge).
The circulation of the blood in the peripheral vessels is favoured / disadvantaged by a low / high stiffness of the arteries. Our study indicates that the stiffness of the arteries, mainly linked to factors such as: age, atherosclerosis, etc., is also modulated by responses to emotional stimuli. This modulation effect is new, or at least rarely reported in the literature. The autonomic nervous system, in response to emotional stimuli, acts through two ways: it changes the rhythm of the heart (HR and HRV) and, at the same time, changes the stiffness of the arteries. This change is temporary and reversible. The stiffness of the arteries is easily detectable, because it modifies the shape of the PPG pulse. So, the shape parameters (SP) are a different biomarker from HRV and PRV, but useful for revealing emotional responses.
Point 4: What are the advantages and disadvantages of PRV measurement compared to HRV. Can we fuse the two signals to improve the performance of emotion recognition?
Response 4: According to our study, there are no advantages in using PRV parameters instead of HRV parameters. If there are, they are very small. The use of shape parameters would seem to improve the performance of the classifier, but this needs to be further investigated. The real, big advantage consists in the use of a sensor, the PPG probe, which is much more convenient, easy to use, non-invasive, compared to the ECG sensor. This advantage is not relevant in laboratory measurements but can be decisive in field measurements: when driving vehicles, when interacting with a computer, when sleeping.
PPG and ECG signals can be merged, but we expect better results from merging with other physiological signals, such as those provided by Galvanic Skin Response (GSR) probes.
Point 5: It is suggested to introduce the data in detail in the form of a table, draw a confusion matrix for the classification results, and conduct cross validation to improve the reliability of the classifier
Response 5: We used the cross-validation procedure (see line 552, page 16) which is, to our knowledge, the most stringent: the LOSO (Leave One Subject Out) procedure. All the values shown in tables 6 and 7 were obtained by applying the LOSO procedure.
Reviewer 3 Report
In this manuscript, the authors have presented the detection of certain physiological parameters related to heartbeat variability using a miniaturized PPG sensor. In addition, the detection rate of the main emotional variables obtained with the PPG technology was compared, and statistically analyzed, with the detection rate obtained using the ECG technology. It is concluded that the results obtained from PPG technique shown similarity with the results obtained from ECG results. The trend of the obtained results was also compared and validated with some previous studies from literature. The novelty presented in this paper was the classification of the parameters using PPG shape parameters. The results were like the other results obtained from standard ECG and PPG techniques presented in this paper. Overall, each section in the manuscript is explained well in details and the manuscript is well drafted. There are a few comments that I have added below:
Comment 1: In Table 3, the SAM score column is not clear. Please explain what does, e.g., 1÷5 mean here when the total intensity levels that were evaluated were 9?
Comment 2: It will be convenient to follow the results section if it is mentioned in detail in the Material and Methods Section that how many groups and subgroups were appointed and which group will be exposed to what procedure? Also, please add the characteristics/differences between each subgroup.
Comment 3: The conclusion of the paper is not clear. The authors claim that PPG is a comparable method to ECG, however, it is not clear that why instead of ECG the proposed PPG method should be used for emotion recognition? What are the advantages of this technique over the standard ECG?
Comment 4: In the conclusion, authors mentioned that this study is merely a first step of a larger project. So, in the future, will the authors be implementing other well-known and more effective machine learning or neural network algorithms besides SVM and KNN methods? It will be interesting to see how the different algorithms could help in the accurate detection.

Author Response
Point 1: In Table 3, the SAM score column is not clear. Please explain what does, e.g., 1÷5 mean here when the total intensity levels that were evaluated were 9?
Response 1: The number of volunteers (31) is not so high as to allow a classification with 9 intensity levels. So, we grouped the 9 levels of intensity (the 5 levels in the case of the familiarity variable) into two classes. Levels 1 to 5 belong to the first class, levels 6 to 9 belong to the second class. I’ll add some words in the text.
Point 2: It will be convenient to follow the results section if it is mentioned in detail in the Material and Methods Section that how many groups and subgroups were appointed and which group will be exposed to what procedure? Also, please add the characteristics/differences between each subgroup.
Response 2: Thank you for the suggestion that allows a better understanding of the procedure used. In section 2.1 Emotion elicitation protocol, the procedures for selecting the final video clips and the two samples used have been better clarified in the text. In this regard, in accordance with the work of Kolestra et al., 2006 (see Koelstra, S., et al. Deap: A database for emotion analysis; using physiological signals. IEEE Trans. Affect. Comput., vol. 3, n. 1, pp. 18-31, 2011) on which we relied, the sample of 40 subjects was used simply to receive a preliminary evaluation of the pre-selected 40 video clips that would allow us to select the best performing ones, the final 8 used for the experimental session here. So, the sample of 40 subjects is independent of the sample of 31 subjects recruited later and who participated in the study presented here.
Point 3: The conclusion of the paper is not clear. The authors claim that PPG is a comparable method to ECG, however, it is not clear that why instead of ECG the proposed PPG method should be used for emotion recognition? What are the advantages of this technique over the standard ECG?
Response 3: According to our study, there are no advantages in using PRV parameters instead of HRV parameters. If there are, they are very small. The real, big advantage consists in the use of a sensor, the PPG probe, which is much more convenient, easy to use, non-invasive, compared to the ECG sensor. This advantage is not relevant in laboratory measurements but can be decisive in field measurements: when driving vehicles, when interacting with a computer, when sleeping.
Point 4: In the conclusion, authors mentioned that this study is merely a first step of a larger project. So, in the future, will the authors be implementing other well-known and more effective machine learning or neural network algorithms besides SVM and KNN methods? It will be interesting to see how the different algorithms could help in the accurate detection.
Response 4: Yes: this study is the first step in a more ambitious project. The next step will involve the use of other non-invasive sensors, such as the GSR, and the fusion of their signals with the PPG signal. Other age groups will also be investigated. The use of more effective algorithms, such as neural networks, is also envisaged.
Round 2
Reviewer 2 Report
Point 1: The method motivation of the paper is missing. It is suggested that the reason for method selection should be explained in the discussion, and the nonlinear HRV measurement and deep learning algorithm should be compared or analyzed to extract the contribution of the paper
Response 1: Thank you for the comment: indeed, the choice of the classification method and its motivation are not sufficiently explained in the text. Our aim was to use a straightforward algorithm to compare parameters extracted from the ECG and PPG waveforms acquisitions. So, we used the most traditional and most popular PRV and HRV parameters. A comparison between several algorithms was not the primary focus of our work. Then we used the most popular algorithms in the field of emotion recognition: SVM and KNN.
SO, The method of the paper is not innovative.....
Point 2: There are large disturbances in HRV measurements, such as individual differences, age / sex. E.g. Robustness Evaluation of Heart Rate Variability Measures for age and gender related autonomic changes in Healthy Volunteers. How to suppress these influencing factors, or what is the difference of HRV measurement between these factors and emotion.
Response 2: Our data does not show a different response to emotional stimuli due to sex. We cannot say anything about the weight of the age factor,....
So, Age, gender, individual differences and other influencing factors exist. The author can refer to relevant literature for analysis, and these factors can be considered more comprehensively in future research. Necessary analysis is needed in the discussion.
Point 3: The main contribution of this paper is not clear. It is suggested to refine the innovation of this paper more clearly through peer comparison (including HRV measurement methods, classification methods, new mechanism discoveries, etc.)
Response 3: The main contribution of our article is the following. We have shown that it is possible to measure the response of the autonomic nervous system to emotional stimuli thanks to a physiological effect, so far not considered or little considered (to our knowledge).
I think that 'it is possible to measure the response of the autonomic nervous system to emotional stimuli thanks to a physiological effect' is not innovative。。。In particular, there are too many studies on autonomic nerve assessment based on ECG or PPG,Relevant literature should be cited for more detailed comparative analysis, and the contributions of this paper are proposed。。。
Point 5: It is suggested to introduce the data in detail in the form of a table, draw a confusion matrix for the classification results, and conduct cross validation to improve the reliability of the classifier
Response 5: We used the cross-validation procedure (see line 552, page 16) which is, to our knowledge, the most stringent: the LOSO (Leave One Subject Out) procedure. All the values shown in tables 6 and 7 were obtained by applying the LOSO procedure.
The table reflects the average results of all models and does not reflect the differences between different models; It is suggested to add a variance of test results of different models, and add a graph to show the difference of cross validation.
